

# Performance of joint modelling of time-to-event data with time-dependent predictors: an assessment based on transition to psychosis data

Hok Pan Yuen[1,2] and Andrew Mackinnon[3,4]

[1] Orygen, The National Centre of Excellence in Youth Mental Health, Parkville, Victoria, Australia
[2] Centre for Youth Mental Health, The University of Melbourne, Parkville, Victoria, Australia
[3] Centre for Mental Health, Melbourne School of Population and Global Health, The University of Melbourne, Parkville, Victoria, Australia
[4] Black Dog Institute and University of New South Wales, Sydney, New South Wales, Australia

## ABSTRACT

Joint modelling has emerged to be a potential tool to analyse data with a time-to-event outcome and longitudinal measurements collected over a series of time points. Joint modelling involves the simultaneous modelling of the two components, namely the time-to-event component and the longitudinal component. The main challenges of joint modelling are the mathematical and computational complexity. Recent advances in joint modelling have seen the emergence of several software packages which have implemented some of the computational requirements to run joint models. These packages have opened the door for more routine use of joint modelling. Through simulations and real data based on transition to psychosis research, we compared joint model analysis of time-to-event outcome with the conventional Cox regression analysis. We also compared a number of packages for fitting joint models. Our results suggest that joint modelling do have advantages over conventional analysis despite its potential complexity. Our results also suggest that the results of analyses may depend on how the methodology is implemented.

## INTRODUCTION

It is quite common in medical research to have a time-to-event variable as an outcome together with time-dependent predictors, which are basically longitudinal data collected on various characteristics. A typical situation is that the outcome is time to death and the time-dependent predictors are biomarkers which may be related to disease progression. Our group work in the youth mental health field in which a highly researched area is the so called ultra high-risk (UHR) patients (*Yung et al., 1996*; *Yung et al., 2003*; *Yung et al., 2004*). These patients are assessed as being at high risk of becoming psychotic. In this scenario, the outcome is time to transition from a non-psychotic state to a psychotic state and the time-dependent predictors are various psychopathological and functioning measures.

Corresponding author
Hok Pan Yuen,
hokpan.yuen@orygen.org.au

The conventional approach used to analyse the kind of data mentioned above is the application of the Cox regression model. However, over the past two decades, methods that can provide a more flexible modelling framework for both the time-to-event and longitudinal aspects have emerged. The resulting models are named joint models. Extensive theory has been developed to provide a high level of flexibility to joint models such as allowing multiple time-dependent predictors and allowing non-parametric, semi-parametric and parametric approaches (*Rizopoulos, 2012b*; *Chi & Ibrahim, 2006*; *Tang, Tang & Pan, 2014*; *Tang & Tang, 2015*). However, the computational requirements for joint models is very demanding. In particular, multiple time-dependent predictors cannot be easily handled by available statistical software. In order to keep the scope manageable, this paper only focuses on the basic joint model with one time-dependent predictor. This paper has two aims. One aim is to assess the performance of joint modelling as compared to the conventional approach. Another aim is to compare the performance of a few joint modelling packages in a situation where the basic joint model is used with specifications that would typically be chosen by users of the respective software. These aims were achieved by simulating data which resembled transition to psychosis studies of UHR patients. The simulated data consisted of the times to transition, a baseline predictor which was labelled as 'group' and a time-dependent predictor whose values corresponded to monthly assessments over 12 months.

## THE COX REGRESSION MODEL

Time-to-event outcomes, such as time to death, time to relapse, time to discharge and so on, are common in medical research. Well established survival analysis methods are commonly used to analyse this type of outcome. Within the context of survival analysis, the Cox regression model (*Kalbfleisch & Prentice, 2002*) has been widely used to seek predictors or risk factors for time-to-event outcomes. The Cox model expresses the logarithm of the hazard rate as a linear function of the predictors. The hazard rate is a measure of the risk of the occurrence of the outcome event. So the Cox model relates the risk to the predictors. The estimated coefficient of each predictor is a measure of the effect of the predictor on the risk. In particular, the exponential value of the estimated coefficient can be regarded as a measure of the hazard ratio for a unit increase in the predictor concerned. There is a major difference between the Cox regression model and some other regression models, which often require an assumption about the underlying probability distribution of the outcome variable. For example, for the linear regression model, we usually need to assume that the outcome measure follows a normal distribution. The beauty about the Cox model is that we do not need to make any assumption about the underlying probability distribution of the outcome data and we can still estimate the effects of the predictors and test hypotheses about the predictors. The way that it achieves this is by using the method called partial likelihood (*Kalbfleisch & Prentice, 2002*; *Cox, 1972*; *Cox, 1975*). This is one of the main reasons why the Cox model is so popular in the analysis of time-to-event data.

The partial likelihood technique can be used when all the predictors are fixed as well as when some or all of the predictors are time-dependent. Fixed predictors are

variables with fixed values for each individual. Examples are gender or age at baseline. Time-dependent predictors are variables with repeated measurements at a number of time points and their values may vary even for a particular individual. The execution of partial likelihood requires the values of the predictors at each recorded event time for all subjects concerned. This does not present any problem for fixed predictors because their values are constant. However, in most circumstances, measurements of time-dependent predictors are made at various assessment time points. Their values are usually not known at event times because the event times do not usually coincide with the assessment time points. A common way to overcome this problem is to employ the last-observation-carried-forward approach and impute the unknown values with the corresponding last recorded values. This approach may not be reasonable if the time gap related to the imputation is large (*Rizopoulos & Takkenberg, 2014*). Also, parameter estimates and standard errors produced by this approach can be biased (*Prentice, 1982*).

## JOINT MODELLING

Research has been carried out in the area of joint modelling since the mid-1990s (*Faucett & Thomas, 1996*) and is still an active research area. The joint modelling technique is designed to tackle the issues mentioned above in the Cox regression model with time-dependent predictors. The working of joint modelling is quite simple conceptually. Considering only one time-dependent predictor, it uses mixed-effects models (*Verbeke & Molenberghs, 2009*) to estimate the trajectory or the trend of the time-dependent predictor and incorporates the estimated trajectory into the Cox regression framework (*Rizopoulos, 2012b*). Specifically, the hazard rate for subject $i$ at time $t$ is given by

$$h_i(t) = h_0(t) \exp(\gamma^T \omega_i + \alpha m_i(t)), \quad i = 1, 2, \ldots, n, t > 0. \tag{1}$$

Here $h_0(t)$ denotes the baseline hazard rate, $\omega_i$ is a vector of baseline predictors (e.g. treatment indicator, gender, age, etc.) and $\gamma$ is the corresponding vector of regression coefficients. The time dependent predictor is represented by $m_i(t)$ with $\alpha$ being the corresponding coefficient vector. A commonly used model for $m_i(t)$ is the linear mixed-effects model. Specifically, $m_i(t)$ is given by:

$$\left. \begin{array}{c} y_i(t) = m_i(t) + \varepsilon_i(t), \\ m_i(t) = X_i^T(t)\beta + Z_i^T(t)b_i, \\ b_i \sim N(0, D), \varepsilon_i(t) \sim N(0, \sigma^2). \end{array} \right\} \tag{2}$$

Here $y_i(t)$ denotes the observed values of the time-dependent predictor for subject $i$, $\varepsilon_i(t)$ is an random error term, $\beta$ is the vector of fixed effects and $b_i$ denotes the vector of random effects with covariance matrix **D**. Both the random errors and the random effects are assumed to be normally distributed. The specifications in Eq. (2) imply that the time-dependent predictor is measured with error and that $m_i(t)$ is the corresponding 'true' value at time $t$. So in joint modelling the association between the event rate and the time-dependent predictor is modelled through the true values of the predictor.

The above specification can be regarded as the basic joint model (*Rizopoulos, 2012b*) and is called the current value parameterization. Many variations to this basic specification of the joint model are possible (*Rizopoulos, 2012b*). For example, interaction terms between the baseline predictors and the time-dependent predictor can be introduced. There can be time-lagged time-dependent predictor, i.e. $m_i(t)$ becomes $m_i(t-c)$ where $c$ is the specified time lag. There can also be time-dependent slope, i.e. the derivative of $m_i(t)$.

Parameters for joint models can be estimated using maximum likelihood (*Henderson, Diggle & Dobson, 2000*; *Hsieh, Tseng & Wang, 2006*; *Wulfsohn & Tsiatis, 1997*) and well-established algorithms such as the Expectation-Maximization (EM) algorithm (*Dempster, Laird & Rubin, 1977*) or the Newton-Raphson algorithm (*Lange, 2004*). Bayesian methods such as MCMC techniques can also be used (*Brown & Ibrahim, 2003*; *Xu & Zeger, 2001*). In this paper, only maximum likelihood estimation is considered.

## SOFTWARE FOR JOINT MODELLING

Joint models can be fitted using the software R, Stata, SAS and WINBUGS (*Gould et al., 2015*). For this paper, the performance of the JM R package Version 1.4-0 (*Rizopoulos, 2012b*; *Rizopoulos, 2010*), the joineR R package Version 1.0-3 (*Philipson, Sousa & Diggle, 2012*; *Philipson et al., 2015*) and the stjm Stata module (*Crowther, Abrams & Lambert, 2013*) were investigated.

The JM package is very versatile and allows many variations to the fitting of joint models. Firstly, it allows the baseline hazard to be unspecified, to take the form of the hazard corresponding to the Weibull distribution for the event times or to be approximated by (user-controlled) piecewise-constant functions or splines. For ordinary Cox regression, the baseline hazard is usually left unspecified. This is of course a well-known advantage of Cox regression. This advantage avoids the restriction resulting from specifying a certain form for the baseline hazard and at the same time still can offer valid statistical inference through the use of partial likelihood. However, in the context of joint modelling, this advantage no longer holds because a completely unspecified baseline hazard will generally lead to underestimation of the standard errors of the model parameters (*Rizopoulos, 2012b*; *Hsieh, Tseng & Wang, 2006*). Although an unspecified baseline hazard function is one of the options in the JM package, the recommendation is that one of the other options should be used.

In addition to the current value parameterization, JM also allows for different specifications of the time-dependent predictors such as time-lagged predictors and time-dependent slopes. It also allows for an accelerated failure time (*Kalbfleisch & Prentice, 2002*) in place of the usual relative risk modelling framework for the survival model. The JM package relies on two other R packages: the survival package (*Therneau & Lumley, 2012*) and the nlme package (*Pinheiro et al., 2012*; *Pinheiro & Bates, 2000*). The former is used to fit the Cox model and the latter is used to fit the linear mixed-effects model. The package JM then extracts all the required information (predictor vectors, design matrices, event indicator, etc.) from the two fitted models to fit the joint model.
The package joineR assumes the Cox proportional hazard model for the time-to-event outcome and it leaves the baseline hazard to be unspecified. The association between the time-to-event and longitudinal components is based on the current value parameterization of the time-dependent predictors. Three options for the specification of the random effects are allowed: random intercept, random intercept and slope, and quadratic random effects. The standard errors of the model parameters are obtained by bootstrap, i.e. by re-estimating the model parameters from simulated realizations of the fitted model. So the above-mentioned issue of underestimated standard errors is not present in joineR. However, the time required to estimate the parameter standard errors can be relatively long because of bootstrapping.

The package stjm allows four options for the specification of the time-to-event outcome. The first three options make use of the proportional hazard model with the baseline hazard derived from the exponential, Weibull or Gompertz distributions. The fourth option utilizes the flexible parametric model, which is based on the cumulative hazard (*Crowther, Abrams & Lambert, 2013*; *Royston & Parmar, 2002*). Three options model the association between the time-to-event and longitudinal components: current value, time-dependent slope and a time-independent structure linking the subject-specific deviation from the mean of the $k$th random effect.

## METHOD

As stated earlier, the Cox model can handle time-dependent predictors. Although it has some potential limitations, its advantage is that it is relatively simple and well established. Joint modelling can potentially overcome the limitations of Cox regression, but it is a much more complicated methodology and is still at a relatively early stage of development. It is of interest to compare the performance of the two. As joint modelling is more demanding computationally, it is also of interest to investigate how the different joint modelling software packages compare. To achieve these aims, a series of simulations were conducted. The details of these simulations are given below.

Each simulation consisted of a time-dependent predictor, a time-to-event outcome and a group variable of two levels. The time-dependent predictor was generated as follows:

i) The timeframe was taken to be days 0, 1, 2, ..., 364.

ii) The value of the time-dependent predictor for subject $i$ at time $t$ ($t = 0, 1, 2, ..., 364$) was generated according to the following linear mixed-effects model:

$$y_i(t) = a_0 + a_1 t + b_{0i} + b_{1i} t + \varepsilon_i(t) \tag{3}$$

iii) $a_0$ and $a_1$ were the fixed effects with given values.

iv) $b_{0i}$ and $b_{1i}$ were the random effects generated from a bivariate normal distribution with mean 0 and a given covariance matrix.

v) $\varepsilon_i(t)$ was the random error generated from a normal distribution with mean 0 and a given variance.

The time-to-event data was generated as follows:

i) The hazard rate, $h_i(t)$, for subject $i$ at time $t$ ($t = 0, 1, 2, \ldots, 364$), was computed as follows:

$$h_i(t) = \exp(\lambda_0 + \lambda_1 m_i(t) + \tau u_i) \tag{4}$$

where $m_i(t)$ denotes the unobserved true value of the time-dependent predictor (i.e. $a_0 + a_1 t + b_{0i} + b_{1i}t$), $\lambda_0$ and $\lambda_1$ are given values, $u_i$ is a group indicator (0 for group 1 and 1 for group 2) and $\tau$ represents the group effect. In the above formulation, $\lambda_1$ is the effect of the time-dependent predictor on survival. More specifically, $\exp(\lambda_1)$ is the hazard ratio for a unit increase of the time-dependent predictor. The baseline hazard is given by $\exp(\lambda_0)$.

ii) Based on Eq. (4), the hazard rate of subject $i$ can vary over time depending on the true value of the time-dependent predictor. But at a particular time $t$, $h_i(t)$ has a particular value and was taken to be the hazard from an exponential distribution. This allowed the generation of the time to event occurrence, $T_i$, for each subject.

iii) The censoring time for each subject, $C_i$, was generated from a uniform distribution on the interval $[1, 364]$.

iv) If $T_i \leq C_i$, the survival status for subject $i$ was taken to be 1 and the time to event occurrence was taken to be $T_i$. Otherwise, the survival status was taken to be 0 and the censoring time was taken to be $C_i$.

To complete the generation of the simulated data, the data collection of the time-dependent predictor was taken to occur at regular time points, specifically at day 0 (i.e. baseline) and then at 30-day intervals thereafter. Also, the data of the time-dependent predictor was taken to be unavailable after the event time or censoring time, whichever was applicable. Therefore, for each subject, non-missing data for the time-dependent predictor were taken to be those at days 0, 30, 60 and so on up to the measurement occasion prior to the event time or censoring time. Any post-event or post-censoring data were not used.

The parameters associated with the simulations were given the following values:

i) The fixed-effect intercept, $a_0$, was given the value 40.

ii) The fixed-effect slope, $a_1$, was given two values, 0.02 and 0.1. These two values were chosen to contrast two different scenarios where one had a steeper trajectory than the other.

iii) The covariance matrix of the random effects, $b_0$ and $b_1$, were given four different forms as shown in Table 1. The four forms were chosen to represent different scenarios of bigger and smaller variances and bigger and smaller correlations for the random effects.

iv) The variance of the random error, $\varepsilon$, was given two values, 16 and 4. The two values were chosen to contrast bigger and smaller error variances.

v) The hazard parameters, $\lambda_0$ and $\lambda_1$, were given the values $-4.8$ and $-0.03$, respectively.

vi) The group effect, $\tau$, was given the value 0. More simulations were done for which a non-zero group effect was used. Details of these simulations are presented later in this paper.

**Table 1 The four different forms used in simulations for the covariance matrix of the random effects, $b_0$ and $b_1$, in Model (3).**

|   | Var($b_0$) | Var($b_1$) | Cov($b_0$, $b_1$) | Correlation |
|---|---|---|---|---|
| a | 32 | 0.002 | 0.06 | 0.237 |
| b | 32 | 0.002 | 0.02 | 0.079 |
| c | 8 | 0.0005 | 0.02 | 0.316 |
| d | 8 | 0.0005 | 0.002 | 0.032 |

Putting these parameter values together, there were 16 different sets of simulations (2 $a_1$ values × 4 random-effect covariance matrices × 2 error variances = 16).

The above simulations pertain to a monotone pattern of data availability for the time-dependent predictor, i.e. the time-dependent predictor is available for each subject from baseline up until the subject is lost (due to event occurrence or censoring). While this pattern would still be of interest for the purpose of simulation, it may not be very realistic. In practice, subjects may miss some of the assessments at individual time-points in a haphazard manner. Also, the actual assessment times in real studies may not always be at fixed intervals due to various practical constraints. Because of these considerations, the above-mentioned 16 sets of simulations were repeated for a non-monotone pattern of data for the time-dependent predictor with irregular assessment times. This pattern was generated as follows:

i) The data of the time-dependent predictor were again generated by the above-mentioned linear mixed-effects model.

ii) Data collection for baseline was taken to always occur at day 0. The nominal time-points for subsequent data collection were again taken to be days 30, 60 and so on. But the actual data collection time-points for each subject were generated by adding a randomly generated quantity to each of the nominal time-points. The quantity was an integer randomly chosen from the interval [−7, 7]. In other words, an assessment window of ±7 days was allowed for each data collection time-point.

iii) The number of post-baseline nominal data collection time-points was 12 (360/30). Of these 12, a randomly selected subset (of size ranging from 0 to 12) was chosen for each subject to contain missing values.

iv) The survival time was generated as mentioned earlier.

v) The censoring time was taken to be the time of the last non-missing assessment unless the generated event time was earlier.

In total 32 sets of simulation (16 × 2) were generated. For each set of simulation, 100 datasets were generated. For each dataset, 150 subjects were generated for each of two groups, i.e. a total of 300 subjects. The R package (version 3.1.1) (*R Core Team, 2014*) was used to produce the simulations.

The data generated from the above simulations resembled a transition to psychosis study in which assessments are done at monthly intervals over a 12-month period

(*Yung et al., 2003*). The group variable represents a fixed factor which can be a baseline characteristic such as gender or a treatment factor in a clinical trial. The simulated values of the time-dependent predictor resembled those of the Global Assessment of Functioning (GAF) Scale (*American Psychiatric Association, 1994*), the baseline values of which was found to be a significant predictor of transition (*Nelson et al., 2013*).

## ANALYSES OF THE SIMULATED DATASETS

As indicated above, joint modelling basically consists of a longitudinal sub-model and a survival sub-model. To analyse the simulated datasets, joint modelling was applied with model (3) as the longitudinal sub-model and model (4) as the survival sub-model. Whenever possible, a non-parametric or semi-parametric approach was adopted for the specification of the baseline hazard in order to avoid the need to choose a particular probability distribution. Also, for the purpose of comparison, Cox regression using only the baseline value of the time-dependent predictor and Cox regression using all non-missing values of the time-dependent predictor were also applied.

Specifically, for each dataset generated, the following analyses were carried out:

i) The R survival package was used to fit Cox regression with the group variable and using only the baseline values of the time-dependent predictor. This was done to provide information as to whether there is any advantage to collect longitudinal measurements when the interest is only on a time-to-event outcome. This can especially be useful when the aim is simply to predict the occurrence of the time-to-event outcome. In other words, we are asking the question: can we just take a simple approach and use the baseline values? This is, in fact, a common approach in transition to psychosis studies probably due to its simplicity (*Nelson et al., 2013*; *Cannon et al., 2008*; *Ruhrmann et al., 2010*).

ii) The R survival package was used to fit Cox regression with the group variable and the time-dependent predictor (using the last observation carried forward approach as indicated earlier).

iii) The joineR R package was used to fit joint model. As mentioned before, the baseline hazard was unspecified in joineR.

iv) The stjm Stata module was used to fit joint model. As stjm does not have the option of leaving the baseline hazard unspecified in the basic joint model, the Weibull distribution was chosen from the distributions available.

v) The JM R package was used to fit joint model with the baseline hazard specified to be a piecewise-constant function, i.e. the baseline hazard was taken to have different values at different time intervals. As mentioned above, a completely unspecified baseline hazard in the context of joint modelling may lead to underestimation of the standard errors. So this semi-parametric approach to the specification of the baseline hazard seeks to avoid the underestimation of the standard errors of the parameter estimates. At the same time, it allows some flexibility to the specification of the baseline hazard without the restriction of choosing a particular distribution.

vi) The JM R package was used to fit joint model with the baseline hazard specified by regression splines this time. Specifically, the log baseline hazard is approximated using B-splines. This is an alternative semi-parametric approach following the same rationale as using a piecewise-constant function.

The JM package offers two options for numerical integration: the standard Gauss-Hermite rule and the pseudo-adaptive Gauss-Hermite rule. It has been shown that the latter can be more effective than the former in the sense that typically fewer quadrature points are required to obtain an approximation error of the same magnitude and computational burden is reduced (*Rizopoulos, 2012b*). So the latter was used in the analyses using JM. For all other options in JM as well as in the other packages, the defaults were used in the analyses.

## RESULTS

The focus of this paper is on the estimation of the effect of the time-dependent predictor and also the group effect on survival. In other words, the focus is on the estimates of $\lambda_1$ and $\tau$, which were given the values of $-0.03$ and $0$, respectively in the simulated datasets. In order to provide a visual presentation of these estimates, Fig. 1 shows these estimates from one of the simulation sets in which the fixed effect slope ($a_1$) is 0.02, the error variance ($\varepsilon$) is 16 and the random effects covariance matrix takes the form: $\text{Var}(b_0) = 32$, $\text{Var}(b_1) = 0.002$ and $\text{Cov}(b_0, b_1) = 0.06$. For this simulation set only, in addition to the analyses mentioned above, the JM package was also used to fit joint model with an unspecified baseline hazard. The purpose of this extra analysis was to gauge the potential underestimation of the parameter standard errors when the baseline hazard is unspecified.

Figure 1A shows the boxplots of the estimates of $\lambda_1$ obtained from the 100 generated datasets. It can be seen that there is substantial bias in the direction of underestimation in the Cox regression using only the baseline value of the time-dependent predictor. A similar but less substantial bias is also seen in the Cox regression using all of the longitudinal values of the time-dependent predictor. A slight bias in the direction of over-estimation also appears in the joint-modelling analysis using the JM package with a piecewise-constant baseline hazard. There appears to be no bias in the results of all of the other joint-model analyses.

Figure 1B shows the boxplots of the standard errors of the estimates of $\lambda_1$. The last boxplot in this figure is for the JM joint model analysis with an unspecified baseline hazard. It demonstrates the substantial underestimation of parameter standard errors for this approach. Figure 1C shows the estimates of $\tau$. All approaches yielded similar estimates which appear to be unbiased. The underestimation of parameter standard errors for the JM analysis with unspecified baseline hazard was not observed for the standard errors of the estimates of $\tau$ (Fig. 1D).

In order to provide a concise summary of the results of the 32 sets of simulations, we focus on the coverage of 95% confidence intervals and the degree of bias in estimation. The latter is expressed as the percentage of estimates less than the true parameter value out

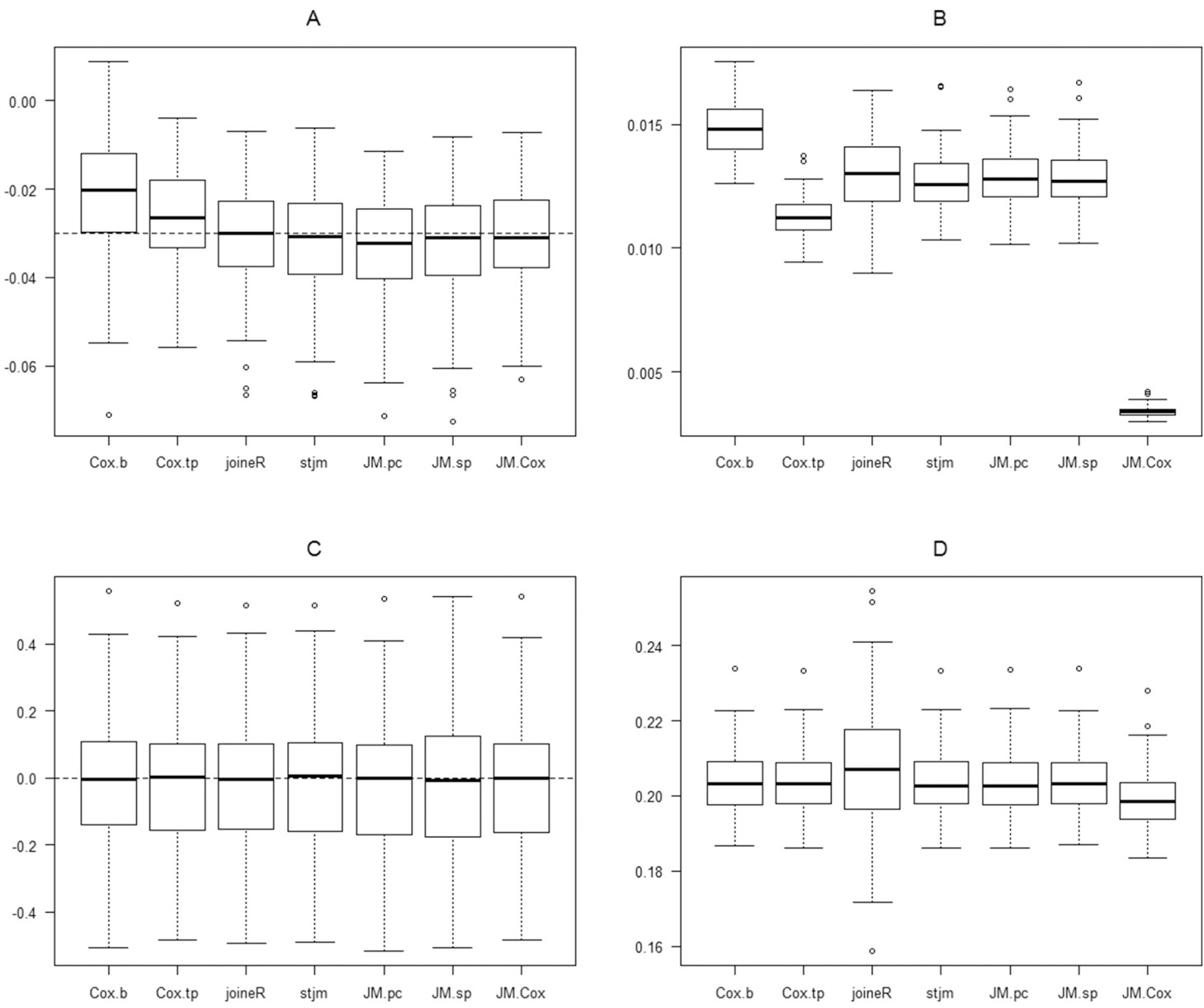

**Figure 1** Boxplots of the results of the analysis of one of the simulation sets in which the fixed effect slope ($a_1$) is 0.02, the error variance ($\varepsilon$) is 16 and the random effects covariance matrix takes the form: $Var(b_0) = 32$, $Var(b_1) = 0.002$ and $Cov(b_0, b_1) = 0.06$. (A) $\lambda_1$ estimates. (B) s.e. ($\lambda_1$ estimates). (C) $\tau$ estimates. (D) s.e. ($\tau$ estimates). $\lambda_1$, Parameter for the effect of the time-dependent predictor on survival (true value = $-0.03$); $\tau$, Parameter for the group effect on survival (true value = 0); Cox.b, Cox regression using only the baseline values of the time-dependent predictor; Cox.tp, Cox regression using all of the longitudinal values of the time-dependent predictor; JM.pc, JM package with a piecewise-constant baseline hazard; JM.sp, JM package with the baseline hazard specified by regression splines; JM.Cox, JM package with an unspecified baseline hazard.

of the 100 estimates for each set of simulation. If the estimation is unbiased, this percentage is expected to be around 50 (as illustrated in the boxplots of Figs. 1A and 1C). Table 2 shows these results for the estimation of $\lambda_1$ in each set of simulation. For the confidence intervals, it is expected that good performance should correspond to a coverage of approximately 95%, say 90% or more. For unbiasedness, as mentioned above, the percentage of estimates less than the true parameter value should be approximately

**Table 2 Results of the estimation of $\lambda_1$ from the 32 sets of simulations for which group effect is zero.**

| m.v. pattern | Var(ε) | $a_1$ | D | Set | Coverage of 95% confidence intervals | | | | | | Percentage of estimates < true value | | | | | |
|---|---|---|---|---|---|---|---|---|---|---|---|---|---|---|---|---|
| | | | | | Cox.b | Cox.tp | joineR | stjm | JM.pc | JM.sp | Cox.b | Cox.tp | joineR | stjm | JM.pc | JM.sp |
| Monotone | 16 | 0.02 | a | 1 | 94 | 94 | 91 | 94 | 97 | 94 | 24 | 40 | 50 | 52 | 59 | 53 |
| | | | b | 2 | 83 | 93 | 92 | 94 | 95 | 95 | 27 | 40 | 52 | 53 | 57 | 52 |
| | | | c | 3 | 82 | 88 | 92 | 93 | 69 | 84 | 20 | 24 | 53 | 58 | 90 | 79 |
| | | | d | 4 | 92 | 90 | 94 | 95 | 91 | 94 | 44 | 24 | 53 | 57 | 40 | 59 |
| | | 0.1 | a | 5 | 88 | 94 | 91 | 96 | 87 | 91 | 33 | 37 | 52 | 56 | 83 | 76 |
| | | | b | 6 | 90 | 94 | 93 | 97 | 88 | 92 | 25 | 32 | 50 | 55 | 85 | 80 |
| | | | c | 7 | 84 | 85 | 94 | 95 | NA | 96 | 19 | 24 | 47 | 58 | NA | 79 |
| | | | d | 8 | 88 | 94 | 91 | 95 | 87 | 91 | 33 | 37 | 52 | 62 | 83 | 76 |
| | 4 | 0.02 | a | 9 | 96 | 95 | 91 | 93 | 96 | 94 | 46 | 52 | 48 | 48 | 56 | 53 |
| | | | b | 10 | 92 | 95 | 93 | 93 | 96 | 95 | 43 | 54 | 53 | 53 | 61 | 53 |
| | | | c | 11 | 92 | 94 | 96 | 97 | 74 | 83 | 39 | 47 | 49 | 50 | 92 | 82 |
| | | | d | 12 | 96 | 95 | 91 | 92 | 96 | 94 | 46 | 52 | 48 | 54 | 56 | 53 |
| | | 0.1 | a | 13 | 92 | 95 | 91 | 95 | 89 | 95 | 48 | 46 | 48 | 54 | 74 | 72 |
| | | | b | 14 | 92 | 96 | 92 | 95 | 88 | 93 | 48 | 49 | 47 | 54 | 80 | 79 |
| | | | c | 15 | 94 | 92 | 94 | 93 | NA | 97 | 35 | 42 | 47 | 60 | NA | 83 |
| | | | d | 16 | 92 | 95 | 91 | 94 | 89 | 95 | 48 | 46 | 48 | 58 | 74 | 72 |
| Non-monotone | 16 | 0.02 | a | 17 | 97 | 96 | 93 | 93 | 96 | 90 | 33 | 31 | 60 | 57 | 44 | 58 |
| | | | b | 18 | 87 | 94 | 88 | 94 | 94 | 93 | 23 | 40 | 49 | 50 | 51 | 53 |
| | | | c | 19 | 85 | 84 | 95 | 95 | 91 | 73 | 21 | 25 | 53 | 53 | 79 | 90 |
| | | | d | 20 | 97 | 96 | 93 | 91 | 96 | 90 | 33 | 31 | 60 | 63 | 44 | 58 |
| | | 0.1 | a | 21 | 95 | 92 | 94 | 95 | 95 | 91 | 33 | 24 | 45 | 55 | 63 | 81 |
| | | | b | 22 | 90 | 88 | 91 | 92 | 95 | 91 | 29 | 21 | 50 | 54 | 62 | 80 |
| | | | c | 23 | 84 | 77 | 94 | 94 | NA | 96 | 20 | 12 | 52 | 63 | NA | 89 |
| | | | d | 24 | 95 | 92 | 94 | 97 | 95 | 91 | 33 | 24 | 45 | 59 | 63 | 81 |
| | 4 | 0.02 | a | 25 | 94 | 95 | 91 | 95 | 96 | 94 | 58 | 61 | 56 | 58 | 57 | 57 |
| | | | b | 26 | 94 | 93 | 92 | 93 | 93 | 93 | 51 | 59 | 52 | 54 | 55 | 55 |
| | | | c | 27 | 92 | 93 | 96 | 96 | 93 | 86 | 43 | 37 | 46 | 47 | 70 | 87 |
| | | | d | 28 | 94 | 95 | 91 | 93 | 96 | 94 | 58 | 61 | 56 | 64 | 57 | 57 |
| | | 0.1 | a | 29 | 96 | 94 | 95 | 93 | 93 | 89 | 63 | 30 | 50 | 52 | 62 | 75 |
| | | | b | 30 | 92 | 96 | 93 | 93 | 94 | 87 | 58 | 40 | 53 | 60 | 63 | 82 |
| | | | c | 31 | 97 | 89 | 96 | 95 | NA | 92 | 46 | 17 | 48 | 58 | NA | 91 |
| | | | d | 32 | 96 | 94 | 95 | 95 | 93 | 89 | 63 | 30 | 50 | 60 | 62 | 75 |

**Notes:**
m.v. pattern, Missing value pattern.
Var(ε), Variance of the random errors in the longitudinal submodel.
$a_1$, Fixed-effect slope in the longitudinal submodel.
D, Covariance matrix of the random effects in the longitudinal submodel. Refer to Table 1.
$\lambda_1$, Parameter for the effect of the time-dependent predictor on survival (true value = −0.03).
NA, Not available due to convergence problems.
Refer to Fig. 1 for other abbreviations.
Shading: 95% Confidence interval coverage < 90% or % estimates < true value is outside the interval [40, 60].

50 for good performance, say between 40 and 60. The shaded entries in Table 2 are those scenarios which did not perform well. It can be seen that joineR and stjm tended to show better results. Note also that, for a small number of the simulated datasets, the

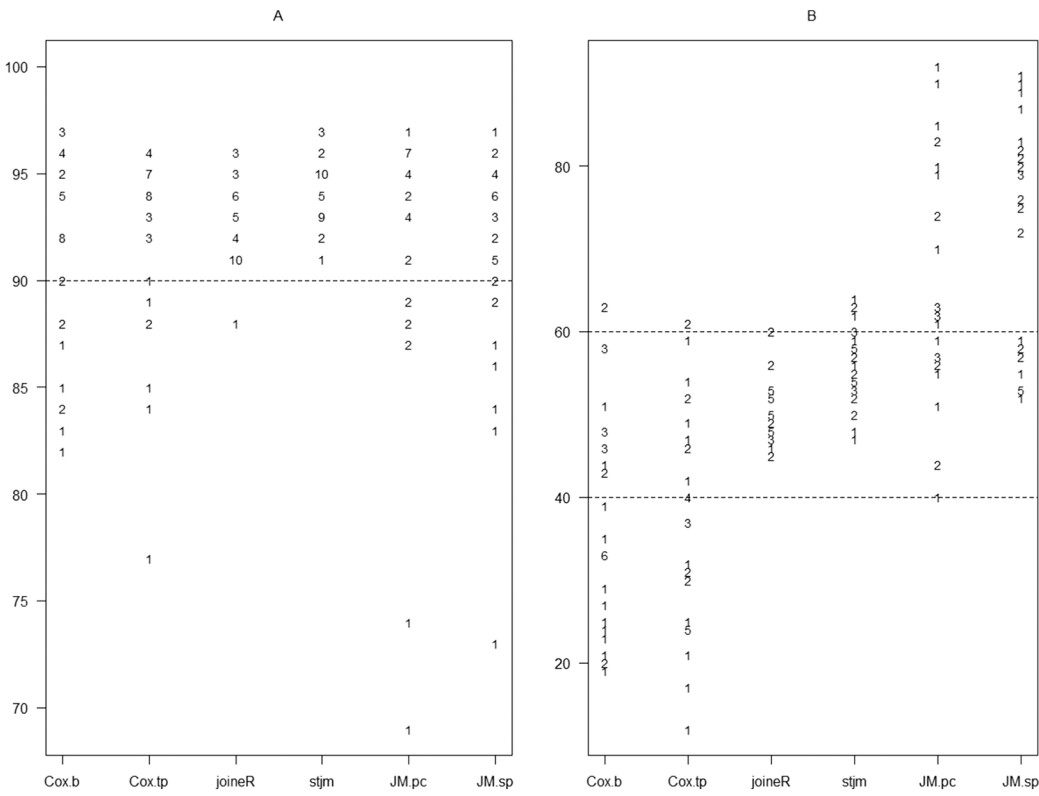

**Figure 2 Graphical presentation of the results in Table 2.** (A) Coverage of 95% confidence intervals. (B) Percentage of estimates less than the true parameter value out of the 100 estimates for each set of simulation. The points are plotted as numbers with each number indicating the number of simulation sets (out of 32) with the corresponding percentage. Refer to Fig. 1 for other abbreviations.

estimates were not available when JM was used with a piecewise-constant baseline hazard due to convergence problems.

The results in Table 2 are further summarized in Fig. 2. In this figure, the points are plotted as numbers with each number indicating the number of simulation sets (out of 32) with the corresponding percentage. For example, the top number under Cox.b in Fig. 2A is 3 with a corresponding percentage of 97. This indicates that 3 of the 32 simulated datasets had a coverage of 97% when 95% confidence intervals were computed for $\lambda_1$ under Cox regression using only the baseline values of the time-dependent predictor. Similarly, the top number under Cox.b in Fig. 2B is 2 with a corresponding percentage of 63. This indicates that 63% of the estimates for $\lambda_1$ were less than the true value for 2 out of the 32 datasets when Cox regression using only baseline values were employed.

From Fig. 2A, it can be seen that all of the datasets had coverage of more than 90% for 95% confidence intervals under stjm. Similarly, all datasets except one had coverage of more than 90% under joineR. For all other analysis methods, quite a number of datasets had coverage less than 90%, with a few much less than 90%.

From Fig. 2B, it can be seen that, for joineR and stjm, nearly all datasets had the percentage of estimates less than the true parameter value between 40 and 60. For Cox regression using only the baseline values and Cox regression using all longitudinal values,
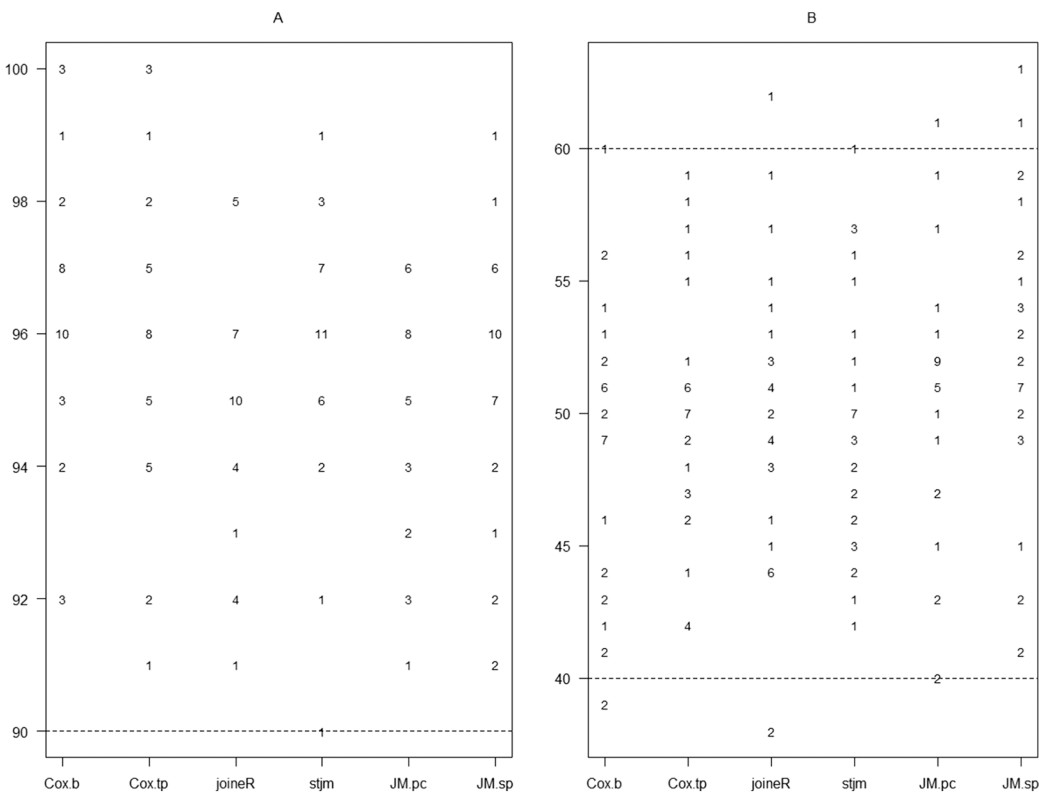

**Figure 3 Graphical presentation of results of the estimation of $\tau$ from the 32 sets of simulations for which $\tau$ is zero.** Refer to Figs. 1 and 2 for the meaning of the labels and the plots.

a substantial number of datasets had the percentage below 40 with some way below 40. As the true value for $\lambda_1$ is negative, this suggests that these two analysis methods tended to underestimate or even reverse the value of $\lambda_1$. Conversely, for the two JM analyses, a substantial number of datasets had the percentage above 60 with some way above 60. This suggests that these two analysis methods tended to overestimate the value of $\lambda_1$.

Figure 3 shows the corresponding results for the estimation of $\tau$. It can be seen from Fig. 3A that all analysis methods had 90% or more for the confidence interval coverage. Figure 3B shows that, except for a few occasions, the percentage of estimates less than the true parameter value were all between 40 and 60 for all the analysis methods. These results suggest that the performance of the different analysis methods were all good for the estimation of the group effect.

Recall that all of the 32 sets of simulations were for a zero group effect. Simulations were repeated on four of the simulation sets for a non-zero group effect. These four sets were sets 3, 7, 19 and 23 in Table 2. These sets were chosen because they showed the worst results for the two Cox analyses and the two JM analyses. The reason that a non-zero group effect was not applied to all the 32 simulation sets was that it was very time consuming to run the simulations. The non-zero group effect was taken to be −0.5. Table 3 shows the results of these simulations, which are very similar to the corresponding results in Table 2.

**Table 3** Results of the estimation of $\lambda_1$ and $\tau$ from four sets of simulations for which group effect is non-zero.

| Estimation of | Set | Coverage of 95% confidence intervals | | | | | | Percentage of estimates < true value | | | | | |
|---|---|---|---|---|---|---|---|---|---|---|---|---|---|
| | | Cox.b | Cox.tp | joineR | stjm | JM.pc | JM.sp | Cox.b | Cox.tp | joineR | stjm | JM.pc | JM.sp |
| $\lambda_1$ | 33 | 84 | 96 | 97 | 97 | 92 | 94 | 11 | 22 | 55 | 61 | 95 | 80 |
| | 34 | 92 | 93 | 97 | 94 | NA | 98 | 23 | 27 | 56 | 59 | NA | 79 |
| | 35 | 83 | 92 | 94 | 96 | 83 | 68 | 20 | 26 | 54 | 58 | 90 | 95 |
| | 36 | 89 | 76 | 91 | 93 | NA | 96 | 19 | 9 | 42 | 58 | NA | 84 |
| $\tau$ | 33 | 97 | 97 | 95 | 97 | 97 | 97 | 52 | 52 | 51 | 53 | 54 | 56 |
| | 34 | 97 | 97 | 94 | 97 | NA | 97 | 52 | 49 | 51 | 50 | NA | 56 |
| | 35 | 97 | 98 | 98 | 98 | 99 | 97 | 54 | 51 | 52 | 52 | 54 | 56 |
| | 36 | 93 | 93 | 94 | 93 | NA | 95 | 49 | 50 | 49 | 50 | NA | 54 |

**Notes:**

$\lambda_1$, Parameter for the effect of the time-dependent predictor on survival (true value = −0.03).

$\tau$, Parameter for the group effect on survival (true value = −0.5).

Sets 33–36 are the same as sets 3, 7, 19, 23, respectively in Table 2 except that the group effect is non-zero.

Refer to Fig. 1 for the abbreviations.

Shading: 95% Confidence interval coverage < 90% or % estimates < true value is outside the interval [40, 60].

## ANALYSIS OF TRANSITION TO PSYCHOSIS STUDY DATA

As a further assessment of the joint modelling methodology and the various software packages, the same analyses described above were applied to a set of real data collected in a study of risk factors of transition to psychosis in which a group of UHR patients were followed up for a maximum period of 12 months (*Yung et al., 2003*). Patients were recruited as they were admitted into the PACE Clinic, which was an outpatient clinical service specifically developed to assess, manage and follow up subjects at high risk of developing a psychotic disorder. Assessments were conducted at study entry and subsequently at approximately monthly intervals until transition to frank psychotic illness occurred, or until 12 months from study entry, whichever came first. For the purpose of illustration, two potential risk factors were considered: family history of mental illness and severity of depression. The former was ascertained as present/absent at entry, indicating whether any first or second degree relatives had mental illness. The latter was measured by the total score of the Hamilton Rating Scale for Depression (HAMD) (*Hamilton, 1960*) which was administered on each occasion of measurement. Scores can range from 0 to 96. Data were available from 47 participants, 29 of whom had a family history of mental illness. The mean HAMD score was 17.7 with an observed range of 2 to 39, representing negligible to severe depression. The number of participants established as having transitioned to psychosis was 21. The time to transition ranged from 7 to 742 days from entry with a mean of 168 days (median 118 days). For those who did not transition, the censoring time ranged from 339 to 707 days with a mean of 450 days. (Transition time and censoring time could exceed 12 months from entry because information from medical records were also used).

The same analytic methods investigated above were applied to the data with family history of mental illness as the group variable and depression score as the time-dependent predictor. The results are shown in Table 4. For the estimation of $\lambda_1$, it can be seen that the estimates can be divided into two groups—those based on the two Cox regressions

**Table 4 Results of the estimation of $\lambda_1$ and $\tau$ from the real dataset.**

| Method | Estimation of $\lambda_1$ | | | Estimation of $\tau$ | | |
|---|---|---|---|---|---|---|
| | Estimate | se | p-value | Estimate | se | p-value |
| Cox.b | 0.065 | 0.0265 | 0.013 | 0.92 | 0.519 | 0.077 |
| Cox.tp | 0.077 | 0.0226 | 0.001 | 0.77 | 0.523 | 0.139 |
| joineR | 0.116 | 0.0425 | 0.006 | 1.00 | 0.600 | 0.095 |
| stjm | 0.128 | 0.0463 | 0.006 | 0.58 | 0.512 | 0.257 |
| JM.pc | 0.118 | 0.0448 | 0.008 | 0.56 | 0.511 | 0.276 |
| JM.sp | 0.129 | 0.0478 | 0.007 | 0.69 | 0.544 | 0.203 |
| JM.Cox | 0.114 | 0.0229 | < 0.00001 | 0.67 | 0.501 | 0.184 |

**Notes:**
$\lambda_1$, Parameter for the effect of the time-dependent predictor on survival (depression score).
$\tau$, Parameter for the group effect on survival (family history of mental illness).
Refer to Fig. 1 for method abbreviations.

and those based on the others. The former are considerably smaller—they are about 60% of the others. This pattern appears to agree with the results of the simulations, which showed that the two Cox regression methods tended to underestimate $\lambda_1$. However, the tendency of the JM analysis to overestimate $\lambda_1$ as observed in the simulations did not appear in this analysis. Neither did the tendency of the JM analysis with unspecified baseline hazard to underestimate the standard error of $\lambda_1$. As the p-values are usually the focus of clinical researchers, it is of interest to note that all the estimates of $\lambda_1$ were significant at the 0.05 level, but the levels of significance varied widely.

For the estimation of $\tau$, while there were also substantial differences between estimates, the standard errors were similar. It is again of interest to note that there was considerable variation among the p-values, although all of them were non-significant at the 0.05 level.

## DISCUSSION

For time-to-event studies with longitudinal predictors, it is conceptually desirable to be able to incorporate the longitudinal data in the prediction of the outcome event. Joint modelling is a welcome addition to the set of analysis tools in such a situation. However, the implementation of joint modelling is computationally challenging (*Rizopoulos, 2012b*; *Wu et al., 2012*; *McCrink, Marshall & Cairns, 2013*). In fact, in the early days of joint modelling, a two-stage approach was suggested (*Self & Pawitan, 1992*; *Tsiatis, Degruttola & Wulfsohn, 1995*). The basic idea of this approach is to firstly estimate the longitudinal trajectories using linear mixed-effects models. Then, the estimates obtained from this first stage are used in the survival models as observed values of the predictors. Strictly speaking, this approach is not joint modelling because the two models are fitted separately. Also, while such an approach is computationally less demanding, the resulting estimates can be biased (*Tsiatis & Davidian, 2001*; *Ye, Lin & Taylor, 2008*; *Ratcliffe, Guo & Ten Have, 2004*; *Sweeting & Thompson, 2011*). The main computational difficulty in joint modelling is that the integrals involved in likelihood estimation are usually intractable in that they have no analytical solutions. So numerical approximations

are usually required to evaluate these integrals. Much research has been done in this aspect (*Henderson, Diggle & Dobson, 2000*; *Wulfsohn & Tsiatis, 1997*; *Song, Davidian & Tsiatis, 2002*; *Rizopoulos, Verbeke & Molenberghs, 2008*; *Rizopoulos, Verbeke & Lesaffre, 2009*; *Rizopoulos, 2012a*). Bayesian methods for the estimation of joint models have also been studied by various authors. But for the purpose of this paper, only estimation based on maximum likelihood is considered.

Much progress has been made in devising techniques to resolve the computational issues involved in joint modelling. As mentioned before, a number of software packages are already available with some of these techniques implemented. However, as joint modelling is still an emerging technique, it is certainly of interest to investigate its performance in practice. *Wu et al. (2012)* conducted a simulation in which the longitudinal predictor followed a linear mixed-effects model with both fixed and random intercepts and slopes. The covariance matrix of the random effects was set to be diagonal and the baseline hazard was set to be constant. They compared the joint-likelihood approach and the two-stage approach and found that the former produced less biased estimates and more reliable standard errors than the latter. *Ibrahim, Chu & Chen (2010)* conducted simulation studies on joint modelling with the longitudinal data generated from a linear mixed-effects model with random effects for an intercept and a slope and also a treatment effect (equivalent to the group effect in this paper). The survival data were generated from the same model as in Eq. (4). They found that, for the treatment effect on survival, both the Cox model with the longitudinal data as a time-dependent covariate and the joint model gave nearly unbiased estimates and gave similar performance in terms of confidence interval coverage. However, for a non-zero effect of the longitudinal data on survival, they found that the former gave biased estimates whereas the latter gave unbiased estimates. Also, the performance of the former was less satisfactory than the latter in terms of confidence interval coverage.

The results presented in this paper broadly agree with these simulation results in the sense that joint modelling could perform better in terms of unbiasedness and confidence interval coverage, especially in terms of the estimation of the effect of the time-dependent predictor on survival. However it seems that the existence of these advantages may depend on how joint modelling is implemented. The joineR package performed very well in our simulations. But its relative weakness at this stage is that it is not very flexible as to how the joint model can be specified. Also, as it utilizes bootstrap to obtain the parameter standard errors, it generally takes much longer to run. For example, for each of the data sets considered in this paper (both simulated and real), joineR took two to three minutes to run; whereas the other packages only took a few seconds. The stjm package also performed quite well, but a potential limitation is that it does not allow a completely unspecified baseline hazard.

Given that joint modelling allows better capture of the information of the changing values of the longitudinal data over time, it would be expected that joint modelling would out-perform the two Cox models considered in this paper especially in a situation when there is much error variation in the longitudinal values. This can be seen in Table 2, which shows that the performance of the two Cox models were especially poor when the error

variance was large ($\text{Var}(\varepsilon) = 16$). In such a situation, joint modelling should certainly be considered.

The analysis of the real data was consistent with the simulations in two respects. The first is that different methodologies and different software may show non-trivial differences in results. The second is that the two Cox-based models do appear to underestimate the parameter of the time-dependent predictor. As this could have a substantive impact on interpretation of effects, it is advisable to use these approaches with caution.

In conclusion, the results in this paper support the conceptual and theoretical notions that joint modelling has the potential to provide better statistical inference when time-to-event outcomes are to be analysed with longitudinal data. The focus of this paper has been on the estimation of the parameters associated with the time-to-event process. But joint modelling can also be utilized to provide predictions for the survival probabilities, estimation for the longitudinal profile as well as prediction for the time-dependent characteristics (*Rizopoulos, 2012b*). So it has the potential to be a powerful and useful statistical tool. Since joint modelling is still a relatively new technique, further development of corresponding statistical software is required. However, existing joint modelling packages have already demonstrated their potential usefulness and should be utilized to apply joint modelling in the analysis of relevant data, subject to the caveats raised in this paper.

### Funding
The authors received no funding for this work. The funders had no role in study design, data collection and analysis, decision to publish, or preparation of the manuscript.

### Competing Interests
The authors declare that they have no competing interests.

### Author Contributions
- Hok Pan Yuen analyzed the data, wrote the paper, prepared figures and/or tables, reviewed drafts of the paper.
- Andrew Mackinnon reviewed drafts of the paper.

### Data Deposition
The raw data has been supplied as Supplemental Dataset Files.

### Supplemental Information
Supplemental information for this article can be found online at http://dx.doi.org/10.7717/peerj.2582#supplemental-information.

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
