# Peer review of "Performance of joint modelling of time-to-event data with time-dependent predictors: an assessment based on transition to psychosis data"

_PeerJ, doi:10.7717/peerj.2582_

## Round 0.1 · original submission · Major Revisions

Dear authors,

Thank you very much for the opportunity to handle your paper.

After reading your work and the reports of the reviewers, I think it has scientific merit to be published in PeerJ, once some issues are solved. Therefore my decision is MAJOR REVISION.

On the other hand, following the comment of the reviewer 1 "1. So far, the joint models of longitudinal data and survival data (JMLSs) have been extended to the context in which multivariate longitudinal data and survival data can be involved, however, multivariate JMLSs are not mentioned in this paper". You may also want to consider reading this paper related to this issue and its application with cardiovascular diseases (Palazón-Bru A, Carbayo-Herencia JA, Vigo MI, Gil-Guillén VF. A method to construct a points system to predict cardiovascular disease considering repeated measures of risk factors. PeerJ. 2016;4:e1673.).

With respect and warm regards,
Dr Palazón-Bru (academic editor for PeerJ)

Reviewer 1 ·

Basic reporting

The authors investigated performance of joint modelling of time-to-event data and time-dependent predictors, compared with conventional Cox regression analysis, also compared a number of packages for fitting joint models and suggested that the results of analyses may depend on how the methodology is implemented. In brief, the paper is generally well written.

I believe that the authors have done a wonderful overview of joint modelling based on comprehendign various method of joint modelling, and that the practical application of joint modelling will be generalized to some extent through comparing the performance of a few joint modelling packages. I really like this paper and believe that it can be published in the journal after some major revisions.

Experimental design

Please see the comments to the authors

Validity of the findings

Please see the comments to the authors

Additional comments

1. So far, the joint models of longitudinal data and survival data (JMLSs) have been extended to the context in which multivariate longitudinal data and survival data can be involved, however, multivariate JMLSs are not mentioned in this paper. Please consider referring to literatures such as [1]-[3].

2. In the lines 116-118 of the paper, it is parametrically specified to model longitudinal data, however, it may obtain misled and biased results under the parametric assumptions. In the literatures [1,2,3], the parametric assumptions have be relaxed to obtain more reasonable analysis, including partially linear functions of covariates on longitudinal responses, normal mixture distribution on random effects and skew normal on random errors. You should consider these nonparametric and semiparametric methods into assessment based on transition to psychosis data for comparison with your proposed methods.

3. Due to Joint models (JM) complexity and intensive computation, it is important to promote the efficiency of various JM packages mentioned in lines 137-139, however, I don’t know the computational burdens of various JM packages, therefore it is necessary to tell how much computing time they take to run a replication in simulations or in the real example.

4. How much is the average censoring rate in simulations and psychosis study data?

5. In the lines 293-300, I wonder how to devide timeframe (time-axis) into different time intervals, especially for such long time span in your simulations, how many different time intervals are needed to obtain good performance?

6. Often in many biomedical and epidemiologic studies, estimating hazards function is of interest, although you introduce different parametric and nonparametric methods to estimate it, the effect to estimate it cannot be found.

Reference
[1]. Chi, Y. Y. and Ibrahim, J. G. (2006). Joint models for multivariate longitudinal and multivariate survival data. Biometrics 62, 432-445.
[2]. Tang, N. S., Tang, A. M. and Pan, D. D. (2014). Semiparametric Bayesian joint models of multivariate longitudinal and survival data. Computational Statistics & Data Analysis 77, 113-129.
[3]. Tang, A. M., & Tang, N. S. (2015). Semiparametric Bayesian inference on skew–normal joint modeling of multivariate longitudinal and survival data.Statistics in medicine, 34(5), 824-843.

Reviewer 2 ·

Basic reporting

Paper is self contained. The structure of the paper is appropriate.

Experimental design

No comment

Validity of the findings

Interpretation needs to be more cautious

Additional comments

Referees report on joint modelling paper by Yuen and Mackinnon, 11005

This paper has three components, a brief review of the concepts behind joint models, a simulation study comparing different software for fitting joint models and illustration of the methods on a specific dataset. The main contribution and the only really new aspect is the simulation study. Comparing different software is a useful service to the community, in that sense the paper makes a contribution.

The paper does have clear findings regarding the relative merits of the various software packages. I have two general comments regarding this. Since you only considered one type of true models, you should be more cautious about extrapolating the relative merits of the softwares to other situations. It is possible that the winning software had distributional assumptions that were more compatible with how the simulated data were generated. The second comment is, can you give any explanation or clues as to why one software did better than another, is it because of more accurate numerical integration, or different optimization algorithms (EM versus Newton-Raphson), or some other reason?

Specific comments.

That paragraph on page 5 about internal and external covariates should be removed. It is not relevant to the paper. Also I don’t agree with some of the statements, eg to say that Cox regression will not work is misleading. The Cox model is about the hazard in the next unit of time, so that is well defined for internal covariates.
In your review of joint models, you should probably credit Faucett and Thomas, who had the first paper on this topic.

Joint models have three uses, although you are restricting your focus to one of them, you should probably mention the others. One use is when the interest is in hazard modelling with a time-dependent covariate, when this covariate is only intermittently measured and has measurement error (this is your use), one is when you are interested in the longitudinal process and the event may cause bias in estimation due to drop-out, and one is for prediction of either the longitudinal or the event time.

Page 7 line 141-142. I don’t think you should say the baseline hazard is approximated by a Weibull distribution. A hazard is not a distribution, do you mean the log hazard is a power of time?

Page 7 lines147-148. You say you get an underestimation of standard errors. But this surely depends on how you calculate the standard errors. Is this based on an information matrix? If so which parameters are included in the information matrix? Would you include the baseline hazard parameters and the longitudinal model parameters in this matrix? What about using the bootstrap?

Are there real references for the joineR software? References 20 and 21 just give dates.

Page 8 for stjm package, the form of the model is not very clear, you talk about proportional hazards, but then you say it is not based on proportional hazards but rather on cumulative hazards.

Page 9 lines 181-182. It is not mathematically demanding. I suggest you simply say it is computationally more demanding.

Page 9. I think you need some notation for the group variable, say Z=0 or 1. Then in equation 4 could be corrected to write tau Z_i instead of tau.

Page 9 line 200. I think you mean unobserved true value, not observed true value.

Page 10 lines 205-207. I don’t understand how T is generated, since h(t) depends on time it can not have an exponential distribution. I think you need to use the cumulative hazard, then use S(t) = exp (- cumulative hazard) and generate from S(t).

Page 14 line 320. Do you mean underestimation of the standard errors, rather than underestimation of the parameter estimates?

In the results section of the simulation I really wanted to learn something about the relative efficiency of the methods, ie the variance of the point estimates. This is an important property of different methods and should be shown, aswell as bias and coverage rates.

Can the results for JM.Cox also be provided.

For the real application the joineR estimate for tau seems out of line with the other methods, it should be noted and an explanation given if you have one.

---

## Round 0.2 · Major Revisions

Dear authors,

After assessing both your revised paper and the reviewers' reports, I think you should make additional changes to your work, as indicated by both reviewers. Therefore, my decision is Major Revisions.

With respect and warm regards,

Dr Palazón-Bru (academic editor for PeerJ)

Reviewer 1 ·

Basic reporting

Although this paper focuses on univariate Joint Model, the authors should point out the current development in this field including multivariate joint models.

Experimental design

The authors did not answer my comments on comment 1 (i.e., there is a paper not cited in the revised manuscript) but they said that they cited it, and comment 2 (i.e., although their focus is on univariate joint models but it is necessary for readers to know the newly development on joint model as I said in the preceding comment 2, it is better for the authors to mention them in Section Introduction and then say the purpose of this article). and comment 5 (i.e. the effect of knots on the results).

Validity of the findings

Please carefully answer the comment 5.

Additional comments

Please carefully consider the above comments.

Reviewer 2 ·

Basic reporting

see below

Experimental design

see below

Validity of the findings

see below

Additional comments

The authors were somewhat, but not completely responsive to my previous comments.
Specific comments.
I continue to think that the paragraph on page 5 about internal and external covariates should be removed, as it doesn’t add anything to the paper, and is a bit misleading. I agree there can be interpretation issues with Cox models for internal covariates, but I think you should remove The statement “the usual likelihood argument is not applicable anymore”. I believe most people would disagree with that. The motivation for using joint models instead of LOCF Cox models is that joint models avoid carrying forward observations and they deal with measurement error, maybe you should make that explicit. Also if you want to make a distinction between internal and external longitudinal variables, then you should be clear on whether joint models can be used for both types of covariates, or just for internal covariates.
Page 6 last line (line 128), remove the word estimated.
Page 10 lines 218-220. You should not use the word exponential for the distribution when the hazard changes with time, by definition an exponential distribution has a constant hazard. I understood the explanation you gave in your response letter, perhaps a brief version of that could be included in the paper. How about saying, the event time was generated from a distribution with piecewise constant hazard rate given by h_i(t).
Page 14 line 309. I think you mean underestimation of the standard errors, rather than underestimation of the parameter estimates?
I believe you misunderstood my previous comment “In the results section of the simulation I really wanted to learn something about the relative efficiency of the methods, ie the variance of the point estimates. This is an important property of different methods and should be shown, aswell as bias and coverage rates.” I wanted to know about the variance (or standard deviation) of the point estimates in the simulations, (this is different from the estimated SE you get from each dataset). Your response letter talked about standard errors, and that was fully explained, but this is a different quantity. If you have saved your simulation results this is easy to obtain.

---

## Round 0.3 · accepted · Accept

Dear authors,

The revised version of your manuscript has high standards to be published in its current form.

Congratulations!

With respect and warm regards,
Dr Palazón-Bru (academic editor for PeerJ)

Reviewer 1 ·

Basic reporting

No Comments

Experimental design

No Comments

Validity of the findings

No Comments

Additional comments

The authors have carefully revised the manuscript accoridng to my previous comments. There is no comments.